# Insights into hydrothermal treatment of biomass blends: Assessing energy yield and ash content for biofuel enhancement

Fidel Vallejo[1]*, Diana Yánez-Sevilla[2], Luis Alonso Díaz-Robles[3], Francisco Cubillos[3], Andrea Espinoza-Pérez[4,5], Lorena Espinoza-Pérez[4,5], Ernesto Pino-Cortés[6], Francisco Cereceda-Balic[7]

1 Industrial Engineering, National University of Chimborazo, Riobamba, Ecuador, 2 Agroindustrial Engineering, National University of Chimborazo, Riobamba, Ecuador, 3 Chemical Engineering Department, Faculty of Engineering, University of Santiago of Chile, Estación Central, Santiago, Chile, 4 Program for the Development of Sustainable Production Systems (PDSPS), Faculty of Engineering, University of Santiago of Chile, Estación Central, Santiago, Chile, 5 Industrial Engineering Department, Faculty of Engineering, University of Santiago of Chile, Estación Central, Santiago, Chile, 6 Escuela de Ingeniería Química, Pontificia Universidad Católica de Valparaíso, Valparaíso, Chile, 7 Centre for Environmental Technologies, Universidad Técnica Federico Santa María, Valparaíso, Chile

☯ These authors contributed equally to this work.
* fidel.vallejo@unach.edu.ec

**Data Availability Statement:** We confirm that our submission contains all the raw data necessary to

## Abstract

This study explores the Hydrothermal Carbonization (HTC) treatment of lignocellulosic biomass blends, delving into the influence of several key parameters: temperature, additive nature and dosage, residence time, and biomass composition. Rapeseeds, Pinus radiata sawdust, oat husks, and pressed olive served as the studied biomasses. One hundred twenty-eight experiments were conducted to assess the effects on mass yield (MY), energy yield (EY), higher heating value (HHV), and final ash content (ASH) by a Factorial Experimental Design. The derived model equations demonstrated a robust fit to the experimental data, averaging an $R^2$ exceeding 0.94, affirming their predictive accuracy. The observed energy yield ranged between 65% and 80%, notably with sawdust and olive blends securing EY levels surpassing 70%, while rapeseed blends exhibited the highest HHV at 25 MJ/kg. Temperature emerged as the most influential factor, resulting in an 11% decrease in MY and a substantial 2.20 MJ/kg increase in HHV. Contrastingly, blend composition and additive presence significantly impacted ASH and EY, with all blends exhibiting increased ASH in the presence of additives. Higher initial hemicellulose and aqueous extractive content in raw biomass correlated proportionally with heightened HHV.

## Introduction

In the context of escalating global energy demand, the pursuit of renewable energy sources has gained significant attention, particularly leveraging available biomass. Biomass, encompassing organic matter from Earth's biological organisms [1], embodies stored solar energy derived

replicate the results of our study, as per PLOS requirements. Also, the raw data is now accessible in the Kaggle repository: 10.34740/kaggle/dsv/8029353.

**Funding:** FV received research funding as part of the ModSim group at the National University of Chimborazo (Universidad Nacional del Chimborazo, Ecuador). The funds were allocated by the Vice Presidency of Research for data analysis, manuscript writing, and covering the Article Processing Charges (APC). The URL is https://www.unach.edu.ec/vicerrectorado-de-investigacion-vinculacion-y-posgrado-ele/. Additionally, research support was provided by BMBF150067 and Dirección de Investigación Científica y Tecnológica, Universidad de Santiago de Chile (DICYT) 062317EP_Ayudante to facilitate data collection and experimental work. The funders had no role in study design, data collection and analysis, decision to publish, or preparation of the manuscript.

**Competing interests:** The authors have declared that no competing interests exist.

through photosynthesis in plant growth. This energy reservoir holds promise for application in combustion processes, and when managed effectively, biomass stands as a renewable and sustainable fuel source [2]. Within this framework, over the past decade, Waste-to-Energy (WtE) technologies have garnered attention for reducing global biomass residue accumulation while unlocking the energy stored within biomass's chemical bonds, yielding gas, fuel, electricity, and other resources [3, 4]. Incineration, once considered a universally simple and efficient method to process biomass residues polluted with other waste streams, presents challenges, as it can yield toxic ash and even dioxins, leading to secondary pollution [5]. Other technologies, such as gasification and torrefaction, have been tested for the valorization of biomass residues. Gasification generates syngas, which presents certain advantages, such as the ability to produce heat simultaneously and serve as raw material for biorefineries. However, tar formation occurs during the process due to incomplete reactions or undesirable re-polymerization of molecule products [6]. Several variables are involved in those mechanisms, one of the main reasons being the composition of the feedstock material [7]. In this context, the moisture content is a critical parameter [8] because the presence of water molecules demands energy for evaporation and chemical reactions, impacting the temperature process and the condensation of higher hydrocarbon compounds.

In summary, a prior drying process of the feedstock material is required, impacting the gasification system's economic cost. A similar situation arises with torrefaction, a process akin to pyrolysis, conducted under inert conditions and temperatures of up to 300˚C. Due to the low reactivity of certain biomass fractions, such as cellulose and lignin, the raw material must undergo size reduction and prior drying processes [9]. In contrast, hydrothermal carbonization (HTC) involves the treatment of materials in subcritical water, and a prior drying stage is unnecessary [10]. The carbon densification achieved through this process can reach up to 50% compared to the raw biomass [11]. The final product, hydrochar, exhibits enhanced calorific values, hydrophobic properties [12], and greater ease of transport and storage [13]. On average, the improvement in calorific values can reach up to 30 to 40% compared to untreated biomass [7]. The low reaction temperature (180–250˚C) and high-moisture biomass conversion without a prior dewatering and drying process are its main advantages over other thermochemical conversion technologies [14]. However, the effects of different input process variables on hydrochar quality are still being studied, aiming for biomass to be effectively used to produce renewable energies [15].

In fact, in the last years, the main goal of the research about HTC was the identification of appropriate biomasses for conversion into hydrochar, considering both economic factors such as abundance, the facility of transportation and storage [16] an energetic perspective, specifically focusing on energy yield [17, 18]. The next challenge of the HTC process development is to achieve high yields on a full scale, complying with quality parameters, notably regarding ash content. This purpose requires evaluating changes in biomass properties, such as mass and energy yields induced by the different operating conditions [19]. Specifically, assessing ash generation in its relationship with additives and the degree of interaction in biomass blends enables finding operational conditions to obtain improved products.

In the case of Chile, in 2018, Chile's energy matrix heavily relied on fossil fuels—coal (20%), natural gas (24%), diesel oil (25%), and firewood combustion (22%)—with only a fraction sourced from renewables. However, given Chile's limited gas, oil, and coal reserves, the nation produced a mere 36% of its annual energy consumption ($4.40 \times 10^{11}$ kWh) domestically, resulting in substantial reliance on imported fuels [20]. This dependence on international markets renders the country's energy security vulnerable to the price volatility of these imported resources. Consequently, the strategic shift aims to elevate non-conventional renewable energies to 70% of Chile's energy matrix by 2050 [21]. Correspondingly, the availability of Chilean

biomass reached 22.40 million m$^3$ by 2021 [13], encompassing residues from sawdust, rapeseed, oats, pressed olive, corn cob, pressed grapes, and other nationally significant sources. However, managing these biomass residues has historically predominantly involved storage in large heaps and landfills and their utilization as fuel in boilers, representing an underexploited reservoir of energy potential.

Some recent research has been reported using HTC for treating Chilean biomasses [22], showing promised results. The reported results indicate an increase in the calorific value of 30 to 40%, keeping the mass yield above 60% [17, 23]. Also, those studies showed the benefit of HTC related to energy valorization of biomass residues for fuel usage but suggested the requirement for optimization of the process [15]. Similarly, challenges highlighted by [24, 25] regarding high ash content in hydrochars suggest optimization.

Regarding some optimization paths for the HTC operating conditions, high variability in biomass composition and its seasonal and geographical availability could be diminished using mixtures, which is crucial for this process [26]. Furthermore, some salts have been proposed as additives to the HTC process to maximize the energy yield. For example, calcium chloride, magnesium lactate, sodium chloride, and lithium chloride [26, 27] improved the energy yield and the higher heating value (HHV) of hydrochar compared to the carbonizations without those additives [26, 28]. However, salts containing mineral components can contribute to ash formation during the HTC process [29]. These minerals, when present in high concentrations, can result in the formation of slag deposits [30].

Moreover, many salts investigated in developing countries have high prices, making using them at full scale impractical [31, 32]. In the case of Chile, Magnesium chloride ($MgCl_2$) could be used in the HTC process due to its lower price and high availability as a by-product of the lithium industry since it has similar physicochemical characteristics to lithium chloride [31, 32].

From this perspective, the novelty of the present work relies on proposing the study of Chilean agricultural and forestry residual biomasses, assessing the influence of operating conditions, including reaction temperature and residence time, in addition to biomass blends and salt additives on improving the hydrochar characteristics focusing on ash content and calorific value. This information could define better operational combinations to produce quality biofuels, mitigate seasonality problems, and contribute to generating non-conventional renewable energies in Chile. Exploring the fate of ash in hydrothermal carbonization (HTC) processes within this context becomes crucial since studies by [33, 34] have shown significant alterations in ash chemistry due to HTC, potentially addressing fouling and slagging concerns.

## Materials and methods

This section details the biomasses utilized, encompassing the characterization of pure biomasses (biomass samples originate from a specific source of biomass residues) and their respective blends. Subsequently, the process conditions were elucidated by outlining the factor levels incorporated within the Experimental Design, including temperature, residence time, type, and dosage of additives. Lastly, the analysis conducted to evaluate the significance of the explanatory variables on the responses is expounded upon.

### Biomass blends

Selected from agricultural and forestry residual sources, the biomasses investigated for conversion to solid fuels included Pinus radiata D. Don sawdust (AS), Brassica napus rapeseed meal (RPS), Avena sativa oat skin (AV), and Olea europaea olive residue (OLV). Notably, 80 to 90%

of Chile residues are typically incinerated, often underutilizing their full energy potential due to combustion in open-air or basic systems.

The biomasses collected from southern Chile were used in their original state for the study. Each experimental run involved carbonizing 240 grams of biomass blends under agitation at 108 rpm and a biomass-to-water ratio of 1:12. Hydrothermal carbonization took place in a high-pressure, high-temperature reactor, specifically the HiPR-SF5L model with a 5 L capacity, equipped with heating surfaces and a thermocouple for internal temperature monitoring. The reactor's pressure setting exceeded the calculated vapor pressure at the wall temperature registered by the thermocouple. The water inlet and outlet were regulated using a butterfly valve, as presented in [35]. Each charge biomass mixture (dry basis) adhered to the weight composition detailed in Table 2. Nitrogen introduction, facilitated through a valve system, displaced air within the reactor chamber, ensuring the HTC process occurred entirely in the liquid phase. All runs maintained a biomass-to-water ratio of 1:12 and agitation at 108 rpm. Anhydrous magnesium chloride (USP, purity > 99%, powdered) and peracetic acid (15 wt. %) were procured from a local Chilean supplier. Moisture (M) content was calculated by the gravimetric method [36], and HHV was determined in a Parr 6200 calorimeter in triplicates with a maximum standard deviation of 150 J/g (Parr Instrument, USA). Ash (A) content was determined according to ISO 18122 [37]. The lignin (L) was determined according to the TAPPI T 222 method [38]. The cellulose (C) and hemicellulose (H) content were obtained using the Kushner-Hoffer and Jaime-Wise methods [39]. Lignin, cellulose, hemicellulose, aqueous extractives (AE), and HHV for each pure raw biomass are presented in Table 1. All reported compositions, and the moisture are in % by weight.

Although biomass residues are widely recognized as a low-cost and readily available raw material for various industrial processes [40, 41], their suitability for specific applications can vary significantly depending on geographical and seasonal factors [42]. Sawdust, for instance, is commonly utilized in producing pellets and briquettes due to its abundance and relatively uniform composition [13]. However, weighing the trade-offs when considering alternative biomass sources such as rapeseed and olive residues is essential. Despite rapeseed and olive residues exhibiting higher heating values (HHV) than sawdust, they also tend to have higher ash content, which can pose challenges in certain applications. Conversely, while presenting a lower HHV than rapeseed and olive, oats boast significantly lower ash content, making them more suitable for applications where ash buildup must be minimized. In light of these considerations, the study investigated the potential synergies in blending different biomass types, with four pairs of biomass blends prepared at a proportion of 70% - 30%. The characterization results of these blends are detailed in Table 2.

**Table 2. Blends characterization.**

| Biomass 1 (70%) | Biomass 2 (30%) | HHV (MJ/kg) | Ash (% dry basis) |
| --- | --- | --- | --- |
| OLV | AS | 21.45 | 2.61 |
| OLV | AV | 20.78 | 0.37 |
| RPS | AS | 20.68 | 4.67 |
| RPS | AV | 20.18 | 5.67 |
| AV | AS | 19.50 | 3.17 |
| AV | RPS | 19.59 | 5.00 |
| AS | RPS | 20.53 | 1.56 |
| AS | OLV | 20.89 | 1.83 |

**Table 1. Raw biomasses characterization.** All the values for a dry basis, AS: Sawdust, AV: Oat skin, RPS: Rapeseed, OLV: Olive, M: moisture, bd: bulk density.

| Biomass | M (%wt) | bd | Proximate analysis (% wt) | | | | | HHV (MJ/kg) | Ultimate analysis (% wt) | | | |
|---------|---------|------|------|------|------|------|------|-------------|-------|------|------|-------|
| | | kg/m³ | A | L | C | H | AE | | C | H | N | O |
| AS | 7.44 | 121 | 0.2 | 30.0 | 42.0 | 25.0 | 2.8 | 20.11 | 51.91 | 6.90 | 0.07 | 40.82 |
| AV | 7.35 | 177 | 4.4 | 12.9 | 37.6 | 23.3 | 21.8 | 18.66 | 46.77 | 7.25 | 0.49 | 40.77 |
| RPS | 7.26 | 617 | 6.3 | 36.3 | 10.3 | 35.3 | 11.9 | 21.34 | 51.68 | 6.97 | 4.97 | 29.43 |
| OLV | 8.20 | 513 | 11.9 | 14.2 | 24.1 | 11.0 | 38.8 | 20.57 | 49.27 | 4.59 | 1.93 | 31.70 |

The results presented in Table 2 provide valuable insights into the effect of biomass blending on the higher heating value and ash content of the resulting mixtures. When comparing biomass blends, it is evident that the combination of OLV with either sawdust AS or oat husks AV consistently yields higher HHV values than other blends. Specifically, the OLV-AS blend exhibits the highest HHV at 21.45 MJ/kg, followed closely by the OLV-AV blend at 20.78 MJ/kg. Conversely, rapeseed press blends yield lower HHV values, with the RPS-AV blend recording the lowest HHV at 19.59 MJ/kg. In terms of ash content, a similar trend is observed, with blends containing RPS showing higher ash content than those with OLV or AS. Notably, the RPS-AV blend has the highest ash content at 5.00%, while the OLV-AV blend exhibits the lowest at 0.37%. These findings suggest that biomass blending can significantly influence the resulting mixtures' energy characteristics and ash content, with OLV-based blends demonstrating promising potential for maximizing HHV while minimizing ash content.

## Experimental design

The goal of the Experimental Design proposed for this study was to evaluate the influence of the nature and dose of additive (D), reaction temperature (T), type of biomass blend (B), and residence time (t) on two process variables: mass yield (MY) and energy yield (EY), and two variables of the final product (hydrochar): ash content (ASH) and calorific value (HHV). The reactor operated under autogenous steam pressure, which was generated during the reaction. While this parameter does not significantly affect the process itself, it's noteworthy that autogenic pressure increases with rising reaction temperature [43]. The factorial design was carried out with two levels for each variable, generating eight sets of 16 experiments, with a total of 128 runs, as shown in Table 3.

Each set allowed evaluation of the influence of replacing the secondary biomass and the impact of the additive. Sets 1–4 indicate the experiments when $MgCl_2$ was used; sets 5–8 were made with peracetic acid.

MY and EY were calculated using Eq (1) and Eq (2), respectively [15]. The significant effects were determined by comparing them with the t-value obtained with 95% confidence for

**Table 3. Experimental design details.**

| SET | T (°C) | t (min) | D | B | |
|-----|--------|---------|---|---|---|
| 1 | 175 | 30 | $1\frac{\text{g MgCl}_2}{\text{g dry biomass}}$ | OLV-AS | OLV-AV |
| 2 | 175 | 60 | | | |
| 3 | 225 | 30 | | RPS-AS | RPS-AV |
| 4 | 225 | 60 | | | |
| 5 | 175 | 30 | $1\frac{\text{mL acid}}{\text{g dry biomass}}$ | AV-AS | AV-RPS |
| 6 | 175 | 60 | | | |
| 7 | 225 | 30 | | AS-RPS | AS-OLV |
| 8 | 225 | 60 | | | |

the two-tailed Student t-test. Indeed, for each set of [24] experiments, the value of t was 2.131, and the variance was calculated using the high-order effects method [44]. Statistical analysis was performed in Statgraphics [45], and the graphs were created using the ggplot2 package [46] in R [47].

$$MY = \frac{\text{Dry hydrochar mass}}{\text{Dry raw biomass mass}} \times 100\% \tag{1}$$

$$EY = MY \times \frac{\text{Hydrochar HHV}}{\text{Raw biomass HHV}} \tag{2}$$

## Results

### Experimental design results

The equations generated by the statistical analysis for each response variable (RV) of mixtures when $MgCl_2$ was used are shown in Table 4. The response equations have been ordered for each main biomass (MB– 70%). The prediction adjustment with the experimental data was satisfactory since an average $R^2$ of 0.95 was calculated, with a minimum value of 0.85. Additionally, the Mean Absolute Percentage Error (MAPE) was 10.3% on average for ASH, with a range of 2.5% to 16.4%, and 1.3% on average for HHV, with a range of 1.0% to 1.8%. All the residuals were assessed with the Anderson-Darling test, demonstrating a normal distribution with 95% confidence, where the value of $A^{2^*}$ was 0.752. These findings indicate that the response equations in Table 4 were appropriately adjusted to the experimental data.

On the other hand, the results obtained for runs with peracetic acid are shown in Table 5. The adjustment obtained for these equations indicates an average $R^2$ of 0.94, with a minimum value of 0.88.

Additionally, the MAPE was 9.8% on average for ASH, with a range of 3.2% to 15.3%, and 0.5% on average for HHV, with a range of 0.3 to 0.7%. All the residuals were assessed with the Anderson-Darling test, indicating a normal distribution with 95% confidence, where the value

**Table 4. Results for sets 1 to 4.**

| MB | RV | Response relation | $R^2$ |
|---|---|---|---|
| OLV | MY | $61.44 - 4.85\,T - 3.09\,B - 1.71\,T\,D + 1.16\,B\,D - 1.02\,t$ | 0.96 |
| | HHV | $24.54 + 1.70\,T - 0.48\,D + 0.435\,T\,D$ | 0.97 |
| | EY | $70.90 - 1.34\,T - 4.20\,B + 1.98\,T\,B - 3.66\,D - 1.30\,T\,D + 3.05\,B\,D$ | 0.96 |
| | ASH | $3.52 - 1.79\,t + 4.17\,D + 1.71\,t\,D - 0.55\,D\,B$ | 0.96 |
| RPS | MY | $54.33 - 5.80\,T - 4.24\,B - 1.76\,T\,D + 1.41\,D$ | 0.95 |
| | HHV | $24.95 + 1.85\,T - 0.50\,D + 0.4\,B + 0.27\,T\,D$ | 0.98 |
| | EY | $65.63 - 4.21\,T - 1.93\,T\,t - 6.55\,B + 1.75\,T\,B - 2.42\,T\,D + 1.86\,B\,D$ | 0.85 |
| | ASH | $3.76 + 0.64\,B + 2.53\,D$ | 0.90 |
| AV | MY | $65.19 - 6.72\,T - 3.08\,B - 2.00\,t + 1.74\,T\,D$ | 0.92 |
| | HHV | $21.20 + 2.15\,T + 0.52\,T\,D - 0.51\,D + 0.50\,B$ | 0.97 |
| | EY | $69.90 - 3.46\,t - 3.57\,B - 2.91\,D + 7.73\,T\,D$ | 0.93 |
| | ASH | $4.82 - 0.54\,t + 0.48\,B + 3.74\,D - 0.49\,t\,D - 0.27\,D\,B$ | 0.99 |
| AS | MY | $73.70 - 10.09\,T - 3.36\,T\,D - 1.72\,D$ | 0.97 |
| | HHV | $22.37 + 1.47\,T + 0.48\,T\,D$ | 0.88 |
| | EY | $78.78 - 11.42\,T - 6.42\,D - 3.94\,T\,D$ | 0.89 |
| | ASH | $3.56 + 2.15\,t + 5.84\,D + 2.10\,t\,D$ | 0.98 |

**Table 5. Results for sets 5 to 8.**

| MB | RV | Response relation | $R^2$ |
|----|----|----|----|
| OLV | MY | $60.95 - 3.01\,T - 3.73\,B + 0.91\,T\,B - 0.67\,D$ | 0.96 |
| | HHV | $24.86 + 1.29\,T + 0.11\,B + 0.18\,t - 0.16\,D$ | 0.99 |
| | EY | $71.49 - 5.80\,B + 2.19\,T\,B - 2.47\,D$ | 0.92 |
| | ASH | $1.91 + 0.44\,t + 0.96\,B + 0.37\,T\,B + 0.95\,D + 0.37\,t\,D$ | 0.90 |
| RPS | MY | $53.33 - 4.32\,B - 4.04\,T + 1.32\,T\,B$ | 0.96 |
| | HHV | $25.36 + 1.55\,T + 0.32\,B + 0.25\,t$ | 0.98 |
| | EY | $65.80 - 1.93\,T - 7.30\,B + 2.45\,T\,B$ | 0.88 |
| | ASH | $2.87 + 0.32\,t + 1.20\,B + 0.53\,t\,B + 0.76\,D + 0.39\,t\,D + 0.43\,B\,D$ | 0.96 |
| AV | MY | $60.82 - 5.35\,T - 4.18\,D - 3.26\,B + 3.12\,T\,D$ | 0.91 |
| | HHV | $21.97 + 1.69\,T + 0.64\,B + 0.28\,T\,B + 0.27\,D + 0.25\,t$ | 0.99 |
| | EY | $67.75 - 3.76\,B - 7.20\,D + 6.56\,T\,D$ | 0.91 |
| | ASH | $3.06 - 0.23\,t + 0.62\,B + 0.23\,D - 0.18\,t\,D - 0.13\,B\,D$ | 0.90 |

of $A^{2*}$ was 0.752. These findings indicate that the response equations in Table 5 were appropriately adjusted to the experimental data.

Although additives increased ashes in all cases, using peracetic acid improved energy efficiency and final calorific value. Besides, the average percentage of ashes is lower. Sets 1–5 and 4–8 indicate significant differences in ash content. The predicted values with the equations have been compared with the experimental values obtained for the process variables (Fig 1) and the properties of the final product (Fig 2).

The legend of the graph indicates by color groups the experiments for each primary biomass and additive used. Besides, an average $R^2$ is shown for each variable, i.e., HHV (0.97), EY (0.91), MY (0.95), and ASH (0.94). It indicates a good fit with the experimental values, and the model equation obtained was enough to explain the variance observed in the results of HTC runs.

The observed variation in the experimental results of MY and HHV was due to the decomposition of the main biomass fractions: cellulose, hemicellulose, and lignin. Although the complete reaction mechanism for this process has not been elucidated, some simplified reaction schemes have been proposed in the literature [48]. The most relevant factor is the temperature

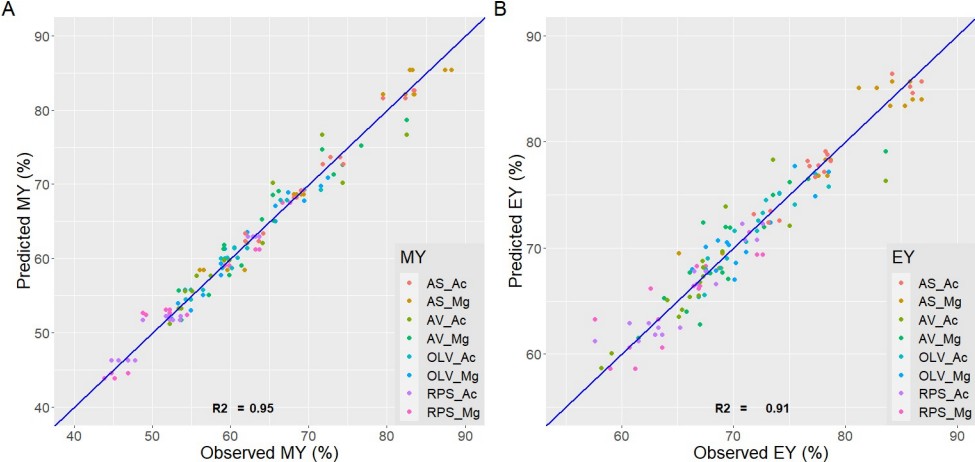

**Fig 1.** (**a**) Predicted and observed MY for all sets; (**b**) Predicted and observed EY for all sets.

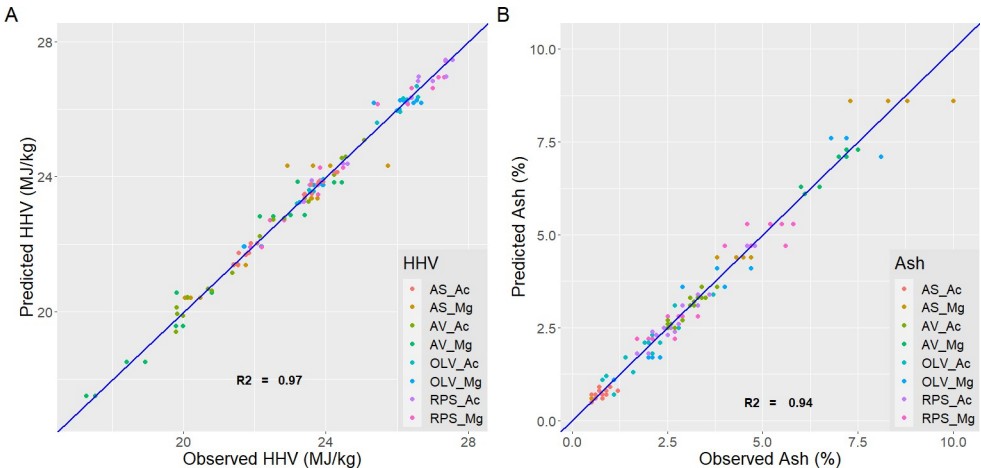

**Fig 2.** (**a**) Predicted and observed HHV for all sets; (**b**) Predicted and observed ASH for all sets.

range in which the reactions of each fraction occur. According to Peterson et al. [49] and based on the kinetic scheme described in Reza et al. [50], hemicellulose reacts from 180°C. Above 230°C, there was no evidence that the biomass contained hemicellulose. On the other hand, cellulose requires more energetic conditions to break its bonds, and it begins to react above 200°C [51]. Finally, lignin and ash content are often considered inert in the usual operational range of HTC, as they have been shown to decompose above 260°C [52]. Another essential concept is the Index of Reactivity (IR), which relates the 'reactive' fraction of biomass (hemicellulose and aqueous extractives) and the 'non-reactive' part (lignin and ash), as shown in Eu 3 [15]. The raw biomass values were used.

$$\text{IR} = \frac{AE + H}{L + A} \tag{3}$$

A higher reactivity index value allows for the obtaining of hydrochar with a higher energy density. Indeed, a recent investigation reported that this factor was significant in developing a linear model for predicting MY and HHV in biomass treated by HTC. Therefore, if biomass has a higher IR, the MY will be lower, and the HHV will be higher under the same time and temperature conditions. This trend can be seen in the experimental results reported in this work.

It is essential to consider the amounts of (H + AE) and (H + C) in the raw biomass, indicated in Table 6.

The studies by Reza et al. (2013) and Vallejo et al. (2020) show that in the range of 150 to 180°C, the aqueous extractives are transferred to the liquid phase completely, while from 180–200°C, the decomposition and polymerization of hemicellulose and cellulose have begun.

**Table 6.** IR, (H+AE) and (H+C) values for raw biomasses.

| Biomass | IR | H+AE (%) | H+C (%) |
|---------|------|----------|---------|
| AS | 0.92 | 27.79 | 67.00 |
| AV | 2.60 | 45.10 | 60.90 |
| RPS | 1.11 | 47.20 | 45.60 |
| OLV | 1.91 | 49.80 | 35.10 |

Therefore, the values determined in the biomass characterization can explain the observed trend at 175 and 225˚C results. The detailed analysis for each variable is described in the following subsections.

## Influence of temperature and time on the response variables

The temperature had the highest impact on the MY and HHV, but for sets 5 and 6, the 'biomass blend' variable had a more significant effect on the MY. A residence time or process temperature increase causes a decrease in MY and an increase in the HHV. Various previous studies have pointed out this trend [11, 53, 54]. The increase in time or temperature is often associated with a higher 'severity' of the process [55]. The most critical variable for analysis in this process is the energy yield because it indicates the amount of net energy that can be recovered from raw biomass. This value depends on the MY and the HHV of hydrochar, as shown in Eq 2. The temperature did not have a significant effect on the EY. Those cases, with the temperature factor in the response equation (set 1, 2, 4, 6, and 8), had a negative effect on the EY value, the same as determined for MY.

According to the described temperature ranges, it was observed that the AS blends had the highest average MY: 73.70% and 71.70% with $MgCl_2$ and peracetic acid, respectively. The lowest H + AE content of the AS of all the biomasses studied was the main reason for this trend.

On the other hand, hemicellulose and cellulose have reacted partially or totally up to 225˚C. The AS had the highest content of both fractions (H+C), and the temperature had a more significant influence than the rest of the blends: 10.10%, as observed in Table 3. On the other hand, an average decrease of 5% was observed for the MY in the OLV blends. It has also been identified that the reduction of MY influences EY more than the increase in HHV. The AS mixtures had the highest values of EY because this biomass had the highest proportion of cellulose and lignin (72%), and the effects of temperature and time were lower than for the other biomass [56]. Besides, the MY of these blends at 175˚C had values close to 90%.

The HHV increase due to the temperature rise was between 1.47 and 2.15 MJ/kg for blends of AS and AV, respectively. The highest values were reached in the RPS mixtures, with an average of 24.95 MJ/kg (+22.3%). AS and RPS are recommended to ensure HHV values higher than 23 MJ/kg with energy efficiencies close to 80%.

In hydrothermal carbonization, the movement of inorganics from the biomass to the liquid phase reduces the hydrochar of minerals and ash formation during combustion while increasing the high heating values [57]. This phenomenon reduces ash content and mitigates fouling and slagging during combustion, improving overall combustion efficiency and exploiting biomass's calorific properties [58]. However, the temperature in HTC significantly influences the behavior of ash and inorganic elements. As the temperature rises, there's an enhanced removal of inorganics such as chlorine, sulfur, nitrogen, and a considerable share of alkali metals, contributing substantially to slagging and fouling issues during combustion [59]. Simultaneously, the residence time in HTC has a significant role in the process. While studies highlight that reaction temperature primarily drives biomass degradation and HHV increase, extending residence time can lead to a maximum HHV increase at specific temperatures [60]. This increase is attributed to removing oxygen and volatile compounds and a parallel rise in carbon content [57].

The observed trends in ash content underscore the complex interplay of various process parameters in HTC [33]. Rather than a direct correlation between temperature and ash content, a nuanced relationship was revealed, challenging conventional assumptions. Surprisingly, temperature variations were found to exert limited influence on ash content across the experimental sets, emphasizing the need for a deeper understanding of underlying mechanisms

governing ash dynamics during HTC; other authors have pointed out the ash study to avoid fouling and agglomeration in operation [61]. Other factors, including residence time, additives, and alterations in secondary biomass composition, influenced ash content outcomes significantly. Reactions promoting biomass decomposition under longer residence times at high temperatures and pressures were observed [62, 63], impacting the carbon content of the initial feedstock. Distinct ash reduction trends were observed in specific experimental sets, notably sets 1, 3, and 7, where time duration emerged as a critical factor driving ash reduction, particularly evident in experiments utilizing agricultural residues such as oat husks (AV) and pressed olive (OLV) as primary biomass sources. These agricultural residues, characterized by relatively high initial ash content, demonstrated consistent trends reflecting the observed ash percentages, suggesting inherent characteristics rendering them more amenable to ash reduction under specific process conditions. Recently, HTC of organic waste produced in biogas conversion processes, where the HHV of the hydrochar range was between 14 and 27 MJ/kg, values similar to those achieved in blends with high cellulose and lignin levels in the present study [64].

Furthermore, the use of acid has been the subject of recent studies, as seen in [65], which used sulfuric acid and 0.3 M acetic acid, obtaining HHV increases of 9 to 20%, which are lower values than those achieved in this study, although the authors' analysis focuses on the reduction of specific radicals in the hydrochar. Similarly, using blends (or co-carbonization) allows for biomass with different initial moisture ranges, which favors the process by reducing the need to add water to the reactor charge. In fact, [66], HHV increase values exceeding 50% are reached but with EY of 60 to 70% because this variable is not analyzed for optimization. Finally, a study conducted with coffee grounds residues indicates the possibility of increasing HHV by up to 46%, without additives, while maintaining high levels of furan reduction that pass into the liquid phase of the process [67].

## Other explanatory effect on response variables

Various studies have been carried out to establish the impact of the macromolecular composition of the biomass during the HTC process [19, 68, 69]. The secondary biomass change (30% in the blends) generated an MY decrease (sets 1,3 and 7) and an HHV increase (sets 5 to 8). As indicated before, the variation in the MY directly influenced the EY, generating a decrease in all the sets. The change in biomass caused an increase in the ash content that responds directly to the original content of each biomass. Variable B was significant in all sets except for 1 and 4. Finally, the biomass change was the variable that appeared less frequently in the equations obtained.

The variable associated with using additives appears most frequently in the response equations. Indeed, it was significant in 25 of the 32 equations generated. The variable D was significant for EY in sets 1, 3, 4, 5, 7, and 8. However, its effect on the response variables indicates that its use is not recommended since it decreased HHV (sets 1, 2, and 5) and increased ashes (all sets). Peracetic acid worked better than $MgCl_2$ to improve the reaction since lower MY and higher HHV were observed, with a difference of approximately 2%, similar to those reported by Lynam et al. [26]. The ash content increase could be explained since although MY decreases and HHV increases, a fraction of the hydrochar is obtained through polymerization reactions [10, 70] that would be affected by the reduction of the pH of the solution [71] or by the saturation of the aqueous medium due to the dissolution of salts [26, 32]

The interactions between variables in the experimental design yielded noteworthy insights into the hydrothermal carbonization (HTC) process. While most interactions were deemed insignificant, the interaction between temperature (T) and dosage (D) (T x D) exhibited a pronounced effect, particularly in runs incorporating $MgCl_2$ as an additive (sets 1 to 4). This

observation underscores the substantial influence of $MgCl_2$ dosage on the process dynamics, with temperature exhibiting a similar trend in these experimental runs. These findings differ from those of other authors who have found that the dosage of chlorine salts has been of little significance in the process [72]. Although it can lead to the transfer of chlorine ions to the process liquor, on the other hand, it allows for a substantial improvement in heat heating value and even a decrease in final ash content, as observed for the blends with oats.

Furthermore, the interaction between temperature and biomass composition (T x B) emerged as significant for energy yield (EY) in specific experimental sets (1, 2, 5, and 6), particularly notable when olive and rapeseed press served as the primary biomass sources. Notably, OLV and RPS, characterized by higher proportions of the 'reactive part' at 49.80% and 47.20%, respectively, exhibited substantial impacts on EY compared to sawdust (AS) with a lower proportion at 27.79%. This disparity in the reactive part content elucidates the differences observed in mass yield (MY) averages, as demonstrated in Tables 5 and 6. Specifically, MY was higher for AS-RPS and AS-OLV blends than OLV-AS and OLV-AV blends. Interestingly, adding biomass with a higher proportion of the reactive part led to a decrease in EY. While some interactions between variables, such as T x B and B x D, resulted in moderate increases in EY under certain conditions (e.g., sets 1, 2, 5, and 6).

## Conclusions

Throughout the last decade, research in Waste-to-Energy (WtE) technologies has aimed to mitigate global biomass residue accumulation while harnessing energy. Despite this focus, the industrial-scale implementation of these technologies remains limited worldwide. Within this context, the HTC process stands poised to address the next challenge: achieving full-scale robust energy yields.

The findings underscore the pivotal role of temperature and biomass composition, revealing their prominence while discounting the perceived influence of additive nature and dosage on process enhancement. Specifically, elevating temperatures led to a 5% to 10% reduction in Mass Yield (MY) and a concurrent increase in Higher Heating Value (HHV) by 1.5 to 2.2 MJ/kg. Transitioning between biomasses with higher reactivity engendered a similar effect on MY and HHV, a significant observation given the geographic and seasonal variability in biomass availability. Notably, energy yields (EY) averaged between 65% and 80%, with sawdust blends yielding the highest values. While temperature did not significantly impact ash content across sets, an overall proportional increase was observed in tandem with other variables, except in sets 1, 3, and 7, where reduced residence time correlated with decreased ash content.

Future endeavors should optimize temperature levels and ratios of biomass blends to fine-tune operational parameters, thereby maximizing hydrochar performance using biomass residues. The path towards effective large-scale implementation necessitates a nuanced examination, encompassing the intricate interplay between agricultural and forestry biomass utilization. A comprehensive understanding of the balance between biomass allocation for food and energy, coupled with addressing land-use implications, environmental impacts, and the substantial investment requisite, is pivotal for propelling biofuel development and establishing sustainable waste-to-energy paradigms at a larger scale. Furthermore, forthcoming research could evaluate minor elements such as K, Cl, Mg, Si, Mn, and ash deformation to provide clues to the kinetics of the HTC process.

## Author Contributions

**Conceptualization:** Fidel Vallejo, Luis Alonso Díaz-Robles, Lorena Espinoza-Pérez, Francisco Cereceda-Balic.

**Data curation:** Fidel Vallejo, Andrea Espinoza-Pérez.

**Formal analysis:** Fidel Vallejo, Luis Alonso Díaz-Robles, Andrea Espinoza-Pérez.

**Funding acquisition:** Luis Alonso Díaz-Robles.

**Investigation:** Fidel Vallejo, Luis Alonso Díaz-Robles.

**Methodology:** Fidel Vallejo.

**Project administration:** Luis Alonso Díaz-Robles.

**Software:** Francisco Cubillos.

**Supervision:** Francisco Cubillos.

**Validation:** Fidel Vallejo, Diana Yánez-Sevilla, Andrea Espinoza-Pérez, Ernesto Pino-Cortés.

**Visualization:** Fidel Vallejo, Andrea Espinoza-Pérez, Ernesto Pino-Cortés.

**Writing – original draft:** Fidel Vallejo, Diana Yánez-Sevilla, Andrea Espinoza-Pérez, Lorena Espinoza-Pérez, Ernesto Pino-Cortés.

**Writing – review & editing:** Fidel Vallejo, Diana Yánez-Sevilla, Lorena Espinoza-Pérez, Ernesto Pino-Cortés.

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
