## [Decision Letter · Decision Letter 0]

12 Mar 2024

PONE-D-23-43899Quantifying Energy Yield and Ash Content in Hydrothermally Treated Lignocellulosic Biomass Blends: Insights for Biofuel OptimizationPLOS ONE

Dear Dr. Vallejo,

Thank you for submitting your manuscript to PLOS ONE. After careful consideration, we feel that it has merit but does not fully meet PLOS ONE’s publication criteria as it currently stands. Therefore, we invite you to submit a revised version of the manuscript that addresses the points raised during the review process.

We look forward to receiving your revised manuscript.

Kind regards,

Nor Adilla Rashidi, Ph.D.

Academic Editor

PLOS ONE

Journal Requirements:

2. Please expand the acronym “DICYT” and "UNACH" so that it states the name of your funders in full.

Reviewers' comments:

Reviewer's Responses to Questions

**Comments to the Author**

1. Is the manuscript technically sound, and do the data support the conclusions?

Reviewer #1: Partly

Reviewer #2: Yes

2. Has the statistical analysis been performed appropriately and rigorously? 

Reviewer #1: Yes

Reviewer #2: Yes

3. Have the authors made all data underlying the findings in their manuscript fully available?

Reviewer #1: Yes

Reviewer #2: Yes

4. Is the manuscript presented in an intelligible fashion and written in standard English?

Reviewer #1: Yes

Reviewer #2: Yes

5. Review Comments to the Author

Reviewer #1: Comments for the manuscript entitled “Quantifying Energy Yield and Ash Content in Hydrothermally Treated Lignocellulosic Biomass Blends: Insights for Biofuel Optimization” (MS No. PONE-D-23-43899)

Introduction

- The sub-heading of Residues revalorization and Related work should be removed from the introduction. The detail for these can be summarized under the head of introduction.

- Line 50: 4.4 ·1011 kWh should be revised as 4.4 x 1011 kWh

- Line 50: ….imported energy, please check it is energy or fuels. Energy means electricity,…., heat.

- Line 51: check the unit of 22.4 million m3. Is it suitable for biomass?

- Line 60: the definition of waste should be provided before. For this case, waste covers the materials from municipal solid waste (MSW) (organic and inorganic components) and biomass (only organic materials)? This is because the incineration is normally applied for MSW. Management of MSW and biomass is different, the authors need to be clear for this issue.

- Line 65: the function of gasification is not for producing heat, but for generating synthesis gas or syngas, which can be applied to produce heat, power and others.

- Line 67: tar formation is not because of biomass moisture, but it is because of another component and reaction. The authors need to be clear for this issue.

- The clear definition of hydrothermal carbonization (HTC) and its advantages are required. It is only in subcritical water?

- The advantages and drawbacks of HTC should be provided, as well as the comparison between torrefaction.

- The effects of some salts on insight characteristics of hydrochar should be summarized, particularly the slagging and fouling effects, and ash meting behavior.

- Effects of minor and major elements for blended biomass during HTC should be summarized.

- The novelty of this work should be clearly indicated.

Materials and methods

- Line 116: what is the definition of pure biomasses?

- The drawing or figure of HTC system is required, as well as its components.

- The size of bulk density of biomass should be reported.

- The blended samples should be done based on dried mass. The water or moisture should be balance before the process. It should be same for all conditions.

- How the authors determine the lignocellulosic components of biomass both before and after HTC.

- The pressure of HTC at each temperature should be provided.

- The authors need to check this “proximate analysis was performed by the Van Soest method”. Is it correct for proximate components?

- The unit of proximate and ultimate analyses should be clearly indicated as % wt. as received basis, dry basis, dry ash free basis….

- The minor elements such as K, Cl, Mg, Si, Mn,…which related to slagging and fouling formation, as well as ash deformation should be determined and reported.

- The number of digits should be two throughout the manuscript.

- Check the summation of each proximate result, and each ultimate result. Is it 100 % or not? If not, what happen?

Results and discussion

- More discussion related to the effects of operating parameters on thermal decomposition of lignocellulosic components is required.

- The change in mass yield of the product is because of thermal decomposition, or dispersion of sample in hot water? The organic or volatile matter in water after HTC should be determined and reported, and discussed.

- The effects of minor elements in biomass samples on thermal decomposition during HTC are required to explain.

- The remaining minor elements (K, Cl, Si, Mn, Mg,…) in biomass after HTC should be reported and discussed.

- The authors did not perform the optimization by RSM or another optimization technique, so the word “optimization” should be removed from the title.

- If the MY is reported as dried mass, is it important to use HHV for calculating EY?

Reviewer #2: Major comment:

1) Please justify why choose factorial design for design of experiment (DOE)? There are many DOE can be used to predict optimum conditions and to evaluate effects of parameters.

2) Please discuss on the error analysis.

3) The authors are advised to discuss on the validation of the DOE. Additional experiments need to be conducted to validate the optimization results.

4) Please compare your findings with literature.

5) Characterization study? Need to discuss on the characterization study.

6. PLOS authors have the option to publish the peer review history of their article (what does this mean?). If published, this will include your full peer review and any attached files.

Reviewer #1: No

Reviewer #2: No

---

## [Author Response · Author response to Decision Letter 0]

14 Apr 2024

Dear Academic Editor and Reviewers,

The attached document details the changes made to address the observations and suggestions provided for the manuscript. We deeply appreciate the time you have dedicated to reviewing the document.

Best regards,

Dr. Fidel Vallejo

---

## [Decision Letter · Decision Letter 1]

7 May 2024

Insights into Hydrothermal Treatment of Biomass Blends: Assessing Energy Yield and Ash Content for Biofuel Enhancement

PONE-D-23-43899R1

Dear Dr. Vallejo,

We’re pleased to inform you that your manuscript has been judged scientifically suitable for publication and will be formally accepted for publication once it meets all outstanding technical requirements.

Kind regards,

Nor Adilla Rashidi, Ph.D.

Academic Editor

PLOS ONE

Additional Editor Comments (optional):

Reviewers' comments:

Reviewer's Responses to Questions

**Comments to the Author**

1. If the authors have adequately addressed your comments raised in a previous round of review and you feel that this manuscript is now acceptable for publication, you may indicate that here to bypass the “Comments to the Author” section, enter your conflict of interest statement in the “Confidential to Editor” section, and submit your "Accept" recommendation.

Reviewer #1: All comments have been addressed

Reviewer #2: All comments have been addressed

2. Is the manuscript technically sound, and do the data support the conclusions?

Reviewer #1: Yes

Reviewer #2: Yes

3. Has the statistical analysis been performed appropriately and rigorously? 

Reviewer #1: N/A

Reviewer #2: Yes

4. Have the authors made all data underlying the findings in their manuscript fully available?

Reviewer #1: Yes

Reviewer #2: Yes

5. Is the manuscript presented in an intelligible fashion and written in standard English?

Reviewer #1: Yes

Reviewer #2: Yes

6. Review Comments to the Author

Reviewer #1: The revised manuscript is ok, but if the authors would like to maintain the unit of biomass as cubic meter (m3) the authors need to indicate the biomass as slurry biomass or liquid biomass, not solid biomass. I do not agree with the solid biomass as an unit of m3.

Reviewer #2: I am satisfied that the revised manuscript has thoroughly addressed the issues raised in the original manuscript. I would advise that it now be accepted for publication.

7. PLOS authors have the option to publish the peer review history of their article (what does this mean?). If published, this will include your full peer review and any attached files.

Reviewer #1: No

Reviewer #2: No

---

## [Editor Report · Acceptance letter]

10 May 2024

PONE-D-23-43899R1 

PLOS ONE

Dear Dr. Vallejo, 

I'm pleased to inform you that your manuscript has been deemed suitable for publication in PLOS ONE. Congratulations! Your manuscript is now being handed over to our production team.

Kind regards, 

on behalf of

Dr. Nor Adilla Rashidi 

Academic Editor

PLOS ONE